# The Devil Is in the Detail—Understanding Divergence between Intention and Implementation of Health Policy for Undocumented Migrants in Thailand

**DOI:** 10.3390/ijerph16061016

**Published:** 2019-03-20

**Authors:** Rapeepong Suphanchaimat, Nareerut Pudpong, Phusit Prakongsai, Weerasak Putthasri, Johanna Hanefeld, Anne Mills

**Affiliations:** 1Bureau of Epidemiology, Department of Disease Control, the Ministry of Public Health, Nonthaburi 11000, Thailand; 2International Health Policy Program (IHPP), the Ministry of Public Health, Nonthaburi 11000, Thailand; nareerut@ihpp.thaigov.net (N.P.); phusit@ihpp.thaigov.net (P.P.); 3National Health Commission Office of Thailand, Nonthaburi 11000, Thailand; weerasak@ihpp.thaigov.net; 4London School of Hygiene and Tropical Medicine (LSHTM) WC1E 7HT, London, UK; johanna.hanefeld@lshtm.ac.uk (J.H.); Anne.Mills@lshtm.ac.uk (A.M.)

**Keywords:** migrants, access to healthcare, health policy, health insurance, policy implementation, Thailand

## Abstract

Migrants’ access to healthcare has attracted attention from policy makers in Thailand for many years. The most relevant policies have been (i) the Health Insurance Card Scheme (HICS) and (ii) the One Stop Service (OSS) registration measure, targeting undocumented migrants from neighbouring countries. This study sought to examine gaps and dissonance between de jure policy intention and de facto implementation through qualitative methods. In-depth interviews with policy makers and local implementers and document reviews of migrant-related laws and regulations were undertaken. Framework analysis with inductive and deductive coding was undertaken. Ranong province was chosen as the study area as it had the largest proportion of migrants. Though the government required undocumented migrants to buy the insurance card and undertake nationality verification (NV) through the OSS, in reality a large number of migrants were left uninsured and the NV made limited progress. Unclear policy messages, bureaucratic hurdles, and inadequate inter-ministerial coordination were key challenges. Some frontline implementers adapted the policies to cope with their routine problems resulting in divergence from the initial policy objectives. The study highlighted that though Thailand has been recognized for its success in expanding insurance coverage to undocumented migrants, there were still unsolved operational challenges. To tackle these, in the short term the government should resolve policy ambiguities and promote inter-ministerial coordination. In the long-term the government should explore the feasibility of facilitating lawful cross-border travel and streamlining health system functions between Thailand and its neighbours.

## 1. Introduction

The volume of global migration has been increasing rapidly. By 2017 there were approximately 258 million international migrants, accounting for 3.4% of the world’s population [1], many of these without access to health services. A number of international agencies, particularly the United Nations (UN), the World Health Organization (WHO), and the International Organization for Migration (IOM) have strongly advocated policies or measures that aim to protect the health of migrants. This is evidenced by a number of World Health Assembly (WHA) resolutions such as WHA60.26, WHA61.17 and WHA70.15 [2]; and the ‘Global Compact for Safe, Orderly and Regular Migration’ recently adopted by UN member states in 2018 [3].

Southeast Asia is one of the most economically dynamic regions in the world, with a large number of migrants travelling within the region, and between the region and the rest of the world [4]. Amongst countries in the Southeast Asia region, Thailand is one of the recipient countries for international migrants. As of 2018, the majority of international migrants in Thailand were workers and their dependents travelling from Cambodia, Lao PDR, and Myanmar, so-called CLM migrants. The accumulated number of CLM migrants living in Thailand each year was around three million [5]. About two thirds of these migrants crossed the border without valid travel documents and were usually recognized as undocumented or irregular migrants. Though Thai immigration law requires these migrants to be deported, the Thai government adopted a leniency measure by allowing undocumented migrants to live and work in the country legally, conditional upon registration with the government [6]. In other words, undocumented migrants need to undergo a legalisation process to acquire legitimate residence and work permits. 

Registration policies for undocumented migrants in Thailand have varied over time, depending on the openness and political attitudes towards migrants in a particular period [6,7]. Suphanchaimat et al. suggested that migrant policies in Thailand could be divided into four eras [8]. The first era was between the early 1900s to the 1990s when nationalism grew in response to the advent of colonialism in Southeast Asia, and became more apparent in the 1970s due to a fear of communism. One response was the special law (Por Wor 337) which removed the Thai nationality of a person born in Thailand to non-Thai parents [9].

The second era, in the early 1990s, was characterised by the country’s increasing industrialised economy. This resulted in a serious shortage of low-skilled labour, particularly in difficult, dangerous, and dirty jobs (e.g., construction work, fishing, and farming). A huge influx of migrants, fleeing from political violence in Myanmar, was a solution to the country’s labour shortage. Instead of deporting these undocumented migrants, the governments initiated registration policies to allow them to legally reside and work in the country within a given period through issuing work and residence permits. Economic interests meant the government turned a blind eye to the illegal status of migrants [10]. In addition, a Memorandum of Understanding (MOU) was agreed between Thailand and CLM nations to recruit low-skilled migrants directly from their country of origin in a lawful manner. However, undocumented migrants still far outnumbered MOU migrants. As of 2018, around 800,000 migrants were recruited through the MOU compared to about 2 million CLM migrants who crossed the border illegally [5]. 

The third era commenced after 2004 when CLM migrants were required to register with the government in order to participate in ‘nationality verification’ (NV). While NV was being processed, these migrants could obtain a temporary identity card from the Ministry of Interior (MOI), and a work permit from the Ministry of Labour (MOL). Once the NV was complete they would receive a ‘temporary passport’ as evidence of their ‘legalised’ status. 

Parallel to the NV the (registered) CLM migrants were required to buy health insurance at a yearly cost of 1300 Baht (US$ 39). The insurance is called the ‘Health Insurance Card Scheme’ (HICS), managed by the Ministry of Public Health (MOPH). The card price was 1300 Baht (US$ 39) between 2004 and 2013, and 2200 Baht (US$ 67) from 2013 onwards [11]. In 2013 the benefit package was expanded to cover antiretroviral treatment (ART) for HIV/AIDS and the MOPH initiated a new insurance scheme for a migrant child no older than 7 years, charging 365 Baht (US$ 11) for one-year coverage [11]. 

The fourth era, which continues to the present time, began in mid-2014 after the military coup. The military government launched a new measure, namely, the ‘One Stop Service’ (OSS), to facilitate the registration of undocumented migrants. Those failing to register with the OSS are subject to deportation [11,12]. The price of a HICS card in the OSS era was reduced for a migrant adult to 1600 Baht (US$ 48) while the price for a migrant child remained unchanged. Figure 1 shows the timeline of HICS changes.

The inclusion of undocumented migrants in the HICS was commended by some global health agencies as showing the country’s effort in extending Universal Health Coverage (UHC) not only to its nationals, but also to people with precarious legal status such as undocumented migrants [4,13].

Nevertheless, there have been a number of implementation challenges that are yet to be solved, as demonstrated by the fact that the number of insured migrants has varied substantially across years despite strong messages from the government that all undocumented migrants ought to be enrolled in the insurance scheme [14]. Though there is some research on migrants in Thailand, most of it focuses on specific medical services for migrant patients, while research to identify discrepancies between policy intention and actual implementation is relatively sparse [15].

Therefore, this study aimed to (i) explain factors that contributed to the formulation of the above migrant health policies, and (ii) examine gaps and dissonance between de jure objectives and de facto implementation of the policies. It is hoped that the findings can help inform policy makers and relevant academics for further improvement of the entire migrant policies in Thailand. Note that the population scope of this research was confined to undocumented CLM migrants and their dependents who are the target population of the HICS and the OSS. Other groups of non-Thai populations (e.g, tourists, expatriates, and refugees in temporary shelters) were excluded. In addition, the policies of interest are (1) the expansion of the HICS benefit package to include ART and the introduction of the insurance card for a migrant child, and (2) the advent of the OSS.

## 2. Methods 

### 2.1. Conceptual Framework and Scope of the Study 

The conceptual framework consisted of two strands: (1) the policy formulation strand (de jure) and (2) the implementation strand (de facto). The policy formulation strand was adapted from the Agenda Setting Theory of Kingdon [16], which suggests that the translation of an idea onto the policy agenda requires a convergence of three streams, namely, (i) a problem stream which refers to an issue being recognised as important and deserving of policymaker attention, (ii) a proposals/policy stream meaning that a solution or policy to the redress the identified problem exists, and (iii) the politics stream referring to political factors around a given issue. Only when these three streams converge, that is, when an issue is recognised as deserving policymaker attention, when a solution is available, and this is politically palatable and feasible, a policy change may occur. The implementation strand drew on Lipsky’s Street-Level Bureaucracy theory [17]. This suggests that what policy makers expect to happen may not always align with reality and rather that actual implementation of policy is shaped more by the perceptions and routines of local implementers or frontline workers, whom Lipsky called ‘street-level bureaucrats’. The theory also proposes that street-level bureaucrats might adapt the public policies to deal with their day-to-day work pressures/problems. Figure 2 lays out the research’s conceptual framework.

### 2.2. Participants and Study Areas

Qualitative methods were used. The period of data collection was October 2014 to September 2015. The study was divided into two levels: (1) the policy formulation level and (2) the implementation level. 

For policy formulation, most of the analyses and data collections were done at the national level. Data collection techniques consisted of in-depth interviews with seven purposively selected policy makers and/or policy-level officials, narrative reviews on migrant-related laws in Thailand, ministerial announcements, and minutes/proceedings from the MOPH. These included reports publicized in the media during the formulation of the policies. For policy implementation, Ranong province was selected as the study site because it represented an area with the largest ratio of (registered) undocumented migrants to Thai citizens, compared to other provinces. The province is located in the southern region of Thailand and has a large diversity of populations (such as Thais, Burmese migrants, and stateless people). Over 95% of migrant workers in the province were from Myanmar. Seafaring, construction, and farm labour were common occupations amongst migrants (Figure 3). 

Eighteen key informants were identified for the implementation strand. Most were healthcare providers from Ranong Provincial Public Health Office and public health facilities in migrant-populated districts (Provincial Hospital X, Kraburi District Hospital Y, and two health centres affiliated to these hospitals). Purposive sampling was used. Also, snowballing sampling was applied until data were saturated. The researchers also interviewed additional informants from outside the MOPH, such as local officers of the MOI and the MOL, NGO representatives, and employers of migrants. About two thirds of the informants were male. Mean age of the informants was approximately 50 years. Key characteristics of informants are presented in Table 1.

Each interview lasted approximately 45–60 min, was audiotaped with consent, and transcribed verbatim. In the fieldwork, the lead researcher served as the main interviewer and was assisted by a co-researcher (WP). All interviews were carried out face-to-face at the interviewees’ workplace, except in some cases (RN_HC2, RN_MOI1, and RN_B1) where telephone interviews were conducted according to the interviewees’ request. 

### 2.3. Data Analysis and Triangulations

Framework analysis was applied, developing both a deductive and inductive coding frame. Deductive coding was used to craft the interview guides and these reflected the conceptual framework. Inductive coding was done where new information emerged from the fieldwork. The interviewers always asked for relevant documents from the respondents as part of methodological triangulation. Once the analysis was complete, the draft final report was circulated to all interviewees to confirm validity of the results. This approach also allowed the researchers to re-check with the respondents as to whether they agreed to have their quotes anonymously presented in the final report.

### 2.4. Ethics Standards

Ethical approval was received from (1) the Institute for the Development of Human Research Protections in Thailand (IHRP 1778/2557), and (2) London School of Hygiene and Tropical Medicine, UK (Ref 8776). All data are kept anonymous. Dissemination of data can be done only for academic interest and individual information cannot be identified. For the interview process, the participants were informed about the study’s objectives and were assured that confidentiality would be strictly kept. The research team always informed participants that it was perfectly acceptable for them to withdraw from the study at any time or to refuse to answer any question if they felt uncomfortable. 

## 3. Results


**The expansion of the HICS benefit package and insurance scheme for migrant children in 2013.**



**● (de jure) Policy formulation**



**○ Problems stream**


The interviewees suggested two main reasons behind the expansion of the HICS benefit package and the instigation of the migrant child insurance in 2013. The first reason related to the curtailment of Global Fund (GF) financial support for ART in Thailand. This point was flagged by three interviewees (PM_01, PM_04, and RN_NGO1). Though the Universal Coverage Scheme (UCS) for Thai citizens had covered ART since 2006, the HICS did not include ART in its benefit package. Before 2013, the Thai government addressed this problem by seeking financial support from the GF, which contributed about 41% of the budget used for HIV prevention activities. The GF was also the main supporter for ART for uninsured migrants in Thailand. In 2011 the GF adopted a new eligibility criterion, meaning that an upper-middle income country would no longer be eligible to submit a new proposal to the GF. The Thai economy had recently passed the upper-middle income benchmark (US$ 4125) [18], and the inclusion of ART into the benefit package of the HICS was considered a feasible solution to replace GF support [19].

The second reason was external pressure from the US, warning that Thailand would be downgraded to the Tier 3 Trafficking Report, the worst amongst all reporting levels. Four interviewees mentioned this point (PM_01, PM_02, PM_03, and PM_04). ‘Tier 3’ meant a country did not make exhaustive efforts to combat trafficking problems. The US government reported that more than 23,000 Cambodian trafficking victims left Thailand each year and the Thai government did not respond adequately to this problem [20]. Thus, the initiation of the migrant child insurance was intended to demonstrate the country’s ‘Corporate Social Responsibility’ (CSR). One of the interviewees (PM_01) mentioned that the CSR idea was widely discussed in many policy dialogues in the MOPH. The expansion of the HICS to migrant children could be used as an excuse, amongst other things, to counter the Trafficking in Persons (TIPS) Report that the government at least had shown some effort to protect vulnerable populations, especially migrant children. Besides, the MOPH made the migrant insurance more open to all non-Thais, not just migrant workers: indeed the term workers does not appear in the Cabinet Resolution of 2013 [21].
“Children and women are potential victims of human trafficking. I am also a member of the White Ribbon (a campaign against violence on women and children). [The interviewee showed the White Ribbon badge to the researcher while interviewing.] . . . That is why we made the 15-January-2013 insurance policy to enable us to insure all migrants in Thailand . . . and the ‘365-Baht’ card is the country’s CSR. . . . And if we take care of them well, once they return home, they will definitely wish to come back to us.”[PM01]

Yet one of the interviewees (PM_02) commented that the initiation of the 365-Baht card policy reflected unsystematic government management of migrant health problems.
“This (the 365-Baht card) shows how the government has brain but no wisdom. How can they say that this is a charitable gift? . . . If the problem is so huge, it should not be CSR . . . it means we must do something systematically.”[PM02]


**○ Politics stream**


Apart from the US threat, domestic politics also played an important role. The Public Health Minister at that time announced that the MOPH was in the process of improving its health system to promote Thailand as a ‘medical tourism and wellness hub’ in Southeast Asia. Though the message did not clearly detail how the MOPH would provide insurance coverage to all non-Thai populations, it indicated that the MOPH would target three subgroups of non-Thais: (1) medical tourists (especially from the Gulf states), (2) foreign tourists, and (3) foreign workers and/or migrants who were not covered by the social health insurance (including CLM migrants) [22]. For the first group, the MOPH would plan to set charges for those entering Thailand to seek medical treatment, and in the meantime, help the neighbouring countries to develop insurance systems in an attempt to encourage patients to receive care in their own countries. Some measures were piloted, such as signing a memorandum of understanding between some Thai hospitals and border hospitals in Myanmar and Cambodia to boost healthcare collaboration, especially in terms of sharing disease alerts and referring emergency or severe cases [23]. For the second group, the government would find measures to promote medical tourism in Thailand. Some measures were proposed, such as exempting the visa requirement for people from six Gulf states who come to Thailand for treatment purposes under the condition that the patients (or tourists) must show an appointment letter issued by a Thai hospital [24]. The HICS was proposed to cover the third group, that is, the Minister announced that the target beneficiaries of the HICS would include all non-Thais who are not covered by the existing social health insurance. This implied that all foreign workers including expatriates were eligible to the HICS. However, the policy was in an early development phase and sparked much public debate. Some doctor groups called for a removal of the minister, accusing him of weakening the public sector in the face of the medical hub policy through supporting the growth of the private sector. The medical hub policy was not the only reason for the protest, but it was combined with other contentious issues, such as interference in the functioning of public health authorities (including the governing body of the UCS), removal of a hardship allowance for rural doctors, and conflict of interest around the Minister’s circles [25]. The minister was in position for only around one and a half years and was removed from office after the coup in May 2014. At the time of writing, there are no concrete government measures to advance the hub policy.


**● (de facto) policy implementation**


Though the 2013 HICS was relatively open to all non-Thais (including undocumented migrants), the number of card holders was quite low. Around a year after the proclamation of the 2200-Baht card, there were only 66,000 card holders, far from its target of 1 million [14]. Four respondents (PM03, RN_RNH1, RN_E2, and RN_E3) mentioned that the high price of the card was one of the most important reasons contributing to the low number of cards purchased. Moreover, there was a problem with the interpretation of the eligibility of the card buyers because the 2013 HICS regulation did not specify the eligible nationality of a buyer. One of the interviewees (PM03) said that European pensioners were attempting to buy the insurance card, creating huge financial risk in some hospitals since some European pensioners had chronic non-communicable diseases (NCD) which required long-term care, and cost of care for those patients greatly exceeded the card price. To further confuse the situation, the term ‘foreigner’ appeared in the card’s English title, but the term ‘tang dao’ (meaning non-Thai citizen, commonly used to refer to CLM migrants) was used in its Thai title (Figure 4).

To resolve this confusion, the MOPH sent an official letter to all facilities in July 2014, asking health facilities to temporarily stop selling the card to ‘farang’ (the lay Thai term referring to Caucasian or white foreigners) and to await further announcements. Yet, so far there has not been any official message from the MOPH informing the health facilities what further action they should take [26]. An interviewee (PM03) suggested that the reason why the MOPH asked all hospitals to pause the selling of the card to western foreigners was not only the confusion in the text, but more importantly, because the policy was designed for vulnerable migrants, not for (those the interviewee considered as) better-off groups, like European pensioners. In contrast, another interviewee (PM06) considered that as long as the term ‘tang dao’ (non-Thai citizen) appeared on the card, it was justified to sell the card to European pensioners.
“When policy makers talked to the public, they said ‘everybody.’ Then, it created a problem. Can a foreign husband of a Thai wife come to buy the card?. . . Those who bought the card are sick foreigners . . . So, we launched a letter telling the hospitals to stop selling the card (to western foreigners).”[PM03]
“[Laughing] Oh! they use the term, ‘farang’ (referring to Caucasian foreigners). The MOPH must answer whether these foreigners are non-Thais in legal terms.”[PM06]

Before 2013 the direction of migrant policies in Thailand appeared to be open to all non-Thai populations despite a dissonance between de jure policy intention and de facto implementation, combined with confusion over who were the target beneficiaries of the policies. This situation became more complicated since 2014 after the arrival of the coup. The later statement demonstrates how migrant policies, particularly the HICS, turned to be stricter than the pre-coup era and how frontline workers responded to such a situation. 


**The advent of the OSS since 2014 and onwards**



**● (de jure) Policy formulation**


The OSS is the most recently launched measure on migrants. Some key attributes of the OSS are as follows [12]. Firstly, it required the MOI, the MOL, and the MOPH to work together in designated places within a province to facilitate the registration process. Secondly, the OSS targeted migrant workers and dependents from CLM nations only. However, it did not specify a definition of ‘dependents’. Thirdly, the essence of the OSS was that an undocumented migrant must (1) register with the MOI to acquire a legitimate residence permit and undertake the NV, (2) be issued with a work permit by the MOL and (3) pay for health insurance. Fourthly, the junta explicitly declared that there would not be any further extension of the OSS after the end of 2014. The OSS aimed to complete the NV process by 31 March 2015. Undocumented migrants and dependents failing to register with the OSS by the deadline would be deported. Lastly, the MOPH reduced the price of the health insurance card from 2200 Baht (US$ 67) to 1600 Baht (US$ 49). 


**○ Problems stream**


In mid-2014, there was a massive exodus of around 200,000 Cambodian migrants returning home. This happened a few weeks after the coup, and caused a loss of more than US$ 1 million per day in cash flow (from reduced wages and remittances) in Thailand and Cambodia because the Cambodian economy could not absorb the labour in a very short period of time [27]. The OSS was then instigated to attract those migrants back to Thailand, as highlighted by some interviewees (PM03, RN_RNH1, and RN_RNH2).
“Initially, this (the OSS) was a measure to pull Cambodian migrant workers back to Thailand. And finally, it expanded to cover all irregular migrants. . . . At that time, many constructions in Thailand; let’s say roads, expressways, and so on, were badly affected.” [PM03]


**○ Politics stream**


The ‘problems stream’ above was closely linked with the ‘politics stream’. Thai political instability occurred in late 2013, triggered by the People’s Democratic Reform Committee (PDRC) protesting against the elected government. The protesters viewed the government as a puppet of the ex-prime minister, Thaksin Shinawatra, who was accused of corruption and damaging the country’s democracy. The turmoil led to the coup d’état in mid-2014. The coup leader claimed that overthrowing the government was necessary to prevent a potential clash between the PDRC and the pro-Shinawatra supporters (Red Shirts). During the turmoil, there were reports that some Red Shirt leaders had hired Cambodian migrants to join the group. This claim coincided with a report by the Cambodian government that the Cambodian prime minister had appointed Thaksin as an economic adviser to Cambodia [28]. A month after the coup the junta broadcast that the military would now strictly regulate the migrant workforce in Thailand. Days after the speech, Cambodian newspapers began reporting a large number of undocumented migrants returning home. Rumours spread rapidly all over the country that undocumented migrants would be severely punished—encouraging a massive voluntary outflow of Cambodian migrants [29]. One respondent commented that this phenomenon reflected mismanagement of migrant policies by the Thai government.
“I am more than happy to see more than 100,000 Cambodian migrants fleeing the country. It makes the government realise that they (migrants) are not voiceless [bangs the table].”.[PM02]


**● (de facto) Policy implementation**


Though the OSS aimed to resolve all problems related to the undocumented immigration status of migrants, its implementation still faced a number of challenges. There were four subthemes emerging from the interviews: (i) delay in the registration process and poor law enforcement, (ii) unclear policy messages, (iii) lack of coordination between public authorities, and (iv) difficulties in managing the insurance for migrant employees.


**○ Delay in the registration process and poor law enforcement**


Though the OSS aimed to complete NV by March 2015, as of early 2015 there were more than 500,000 registered migrants who had not completed the NV—let alone those who had failed to register with the OSS from the outset. The government subsequently announced an extension of the NV for another 3 months [30]. All interviewees mentioned that they were aware of the existence of unregistered undocumented migrants in the study area. Three respondents (RN_MOI1, RN_WP1, and RN_NGO2) pointed out that not all migrant-related laws/measures were strictly enforced. One of the obvious examples was the zoning policy established locally in Ranong province, where prosecutors were more relaxed around the fish docks where migrant communities were concentrated, but stricter around the city centre. The respondent RN_MOI1 mentioned that based on his experience, if the deportation law had been strictly enforced, there might have been conflicts between the prosecutors and local entrepreneurs—thus the zoning system was reasonable in his view.
“In our area, we tried to block the influx of migrants. But we admit that we still face some limitations. In many work sectors, if we always caught undocumented migrants, there might not be enough workers left. Then, we might have problems with the entrepreneurs. So we need to use other measures aside from law enforcement. For example, we tried to create the zoning area that we will be somewhat strict in the inner city and will be more relaxed in the outer zone.” [RN_MOI1]


**○ Unclear policy message**


Equivocal policy messages lead to confusion at the implementation level as evidenced by the following examples: (1) difference between how the MOPH and the MOI defined the term ‘dependents’, and (2) whether or not an undocumented CLM migrant was able to buy the card if he/she failed to register with the OSS. 

Since the junta did not define dependents, this allowed a variety of interpretations by the authorities. In literal terms, a migrant child aged less than seven was eligible to buy the card at the cost of 365 Baht, while a child aged between eight and 15 was required to buy the card at the same price as an adult. However, to buy the adult insurance card a buyer must first hold a work permit, yet a child under 15 years of age cannot be issued with a work permit according to the Labour Protection Act [31]. Two interviewees (RN_RNH2 and RN_RNH3) at Provincial Hospital X stated that given such confusion, the hospital stopped selling the card to children aged between eight and 15 and was awaiting clarification from the central government.
“Now we are selling the card to only those below seven. For those between eight and 15, we have not opened (the card selling policy) yet. Because the term ‘dependent’ of the MOI uses the cut-off at 15. (Interviewer: So far, is there any consensus for this difference?) No! we have stopped selling the card (to children 8–15 years old) at this moment.” [RN_RNH3]

Another troubling example was whether or not the former health insurance card (2200-Baht card) endorsed before the OSS was still in effect. All healthcare provider interviewees perceived that, after the end of the OSS re-registration in late 2014, undocumented migrants were not allowed to buy the insurance card although there had been no official message from the MOPH informing local health facilities to stop selling the 2200-Baht card. This interpretation was in accordance with the political atmosphere at that time as the junta repeatedly informed the media that undocumented migrants who failed to register with the OSS would be arrested. Yet some interviewees (PM06 and RN_NGO2) mentioned that undocumented migrants not joining the OSS were still eligible to buy the 2200-Baht card as long as there was no cabinet resolution to revoke this policy. One of the respondents (RN_PHO1) shared his experience in voicing his concern to the MOPH. The answer from the MOPH was unclear and the MOPH even informed ground-level providers to decide what they deemed appropriate themselves.
“(Interviewer: If I were Burmese, and I somehow did not join the One Stop Service, what would you do to me?) We dare not sell the card. Suppose we sell, there might be a question whether we are against the national policy. (Interviewer: Have you ever raised this issue to the MOPH?) I did. Dr XXX (policy maker in the MOPH) told me that ‘Yes! you may sell them the card but do this covertly.’ I then replied, ‘Sir! If you say so, no local facility will dare sell the card.’ [RN_PHO1]


**○ Lack of coordination between public authorities**


Not only was there confusion about policy content, how the policy was communicated was also a critical problem as raised by six interviewees (RN_PHO1, RN_PHO2, RN_RNH1, RN_RNH2, RN_MOI1, and RN_WP1). The interviewee from the MOI (RN_MOI1) exemplified this point. Since the OSS was quickly endorsed and related authorities were not well informed how to operate the measure in detail, many constraints arose, including a debate about who should absorb the cost of setting up the OSS.
“There were some legal and administrative constraints re[garding] the reimbursement of extra stipend for staff or the problem about human shortage. Because when you summoned lots of staff in a short time to work in a special venue, you needed to ask for help from many authorities. But it is difficult for us (the MOI), as the host (of the venue) to ask for support from others. Because if we cannot give them an extra stipend, they might ask why they have to participate in this event (the OSS).” [RN_MOI1]

Two interviewees (RN_RNH1 and RN_RNH2) from the MOPH reported that they felt that the health sector was ‘voiceless’, and the feedback mechanism from the ground level to the central authorities was also lacking.
“(Interviewer: Could you please tell me about the coordination between you and non-MOPH authorities?) Frankly, we are voiceless. The two parties (the MOI and the MOL) will inform us after they had already talked to, and agreed with each after.” [RN_RNH2]

Conflict between ministries was derived from not only a lack of cooperation, but also a misunderstanding of roles/responsibilities between authorities. An instance was drawn from the argument between the MOPH and the MOI. Though the MOPH intended to have all registered migrants buy the insurance card, the MOPH did not have any legal power to penalise migrants, or employers of migrants, who refused to buy the card because literally, the HICS was just a ministerial announcement.
“For example, the MOPH always told us to force everybody to buy the insurance. But if they could not afford the price, can we force them (to buy the card)? To my knowledge, it is just a ministerial announcement. The MOPH told us to speak in the same language (that all migrants are obliged to buy the card). That makes us feel uncomfortable (to say so).” [RN_MOI1]


**○ Difficulties in managing the insurance for migrant employees**


All four employers (RN_E1, RN_E2, RN_E3, and RN_B1) stated that the HICS created difficulties for employers and should not be compulsory. Their rationale was that most migrant workers, especially those working in offshore fishing boats, were very mobile. Besides, most seamen spent much of the time offshore. Thus, the employers considered that it was not worth paying for the insurance for their employees as they had fewer chances to use services. This problem was coupled with the registration of migrant workers. Legalised migrants (those who passed the NV) were able to travel throughout the country. From the perspective of employers, this regulation created the risk of losing their employees. In contrast, undocumented migrants, who had not completed the NV, were not allowed to move outside the registered province. As a result, it appeared that migrant employers were not supportive of the registration measures, including the health insurance card policy and the OSS.
“I always opposed the HICS. If that is for land migrants or those at the fish docks, I will be OK with it. But for seafarers, I totally disagree because they don’t have a chance to use the insurance . . . I lost over a million for the insurance. Some migrants stayed with me for just a couple of months, then they left their work. And who paid for their insurance? It is the employer! . . . Do you think this policy is successfully implemented? I think it was just 30% successful.” [RN_E3]

Another problem raised by the interviewees was the red tape in the registration. Two employers (RN_B1 and RN_E2) pointed out that the registration process was burdensome. Use of private intermediaries (or brokers) was considered an effective means to overcome this difficulty despite causing additional expenses.
“Now there emerges a new service that helps complete the registration for migrants on behalf of the employers . . . it is more convenient but I had to pay more (laugh!). It charged me 500 Baht per head of migrant. But the registration takes numerous steps, and is very tiresome, and there are so many people. That’s why I don’t want to get involved. So I am OK with hiring them (brokers).”[RN_E2]

With the accounts above, the discrepancies between policy intention and policy implementation can be summarised in Table 2 below.

## 4. Discussion

Thailand is amongst the few developing countries that have achieved UHC. Also, the country has made significant progress in extending UHC to its non-Thai populations, particularly through the introduction of the HICS for undocumented migrants. This achievement was widely recognized by media and academics around the world. For instance, in 2016 National Public Radio in Washington DC published an article on its webpage praising Thailand as the ‘only’ country that offers UHC to all migrants [32]. Likewise, Upneja [33] mentioned in the Yale Global Health Review that Thailand is a model for migrant healthcare for other countries. 

Nevertheless, this study clearly revealed that there is ‘devil in the detail’. It is not argued that the Thai migrant health policies are a failure, but rather that if policy makers wish to further improve migrant health policies, it is crucial to learn from the past as well as the existing situation—how the policies towards migrant health in Thailand were formulated and implemented and what the implementation challenges are. 

From the policy formulation angle, using Kingdon’s Agenda Setting Theory [16], the formulation of the HICS and the OSS was strongly influenced by pressures from international and Thai politics. Examples are, among other things, the pressure from the US on Thailand regarding insufficient effort to combat human trafficking, the weaning off GF support for ART for migrants, and the exodus of Cambodian migrants, which contributed to the OSS in 2014. On the one hand, the Thai government may be seen as responsive to emerging problems by further improving initiatives to protect the health of migrants when ‘windows of opportunities’ opened. On the other hand, their reactions reflect a lack of long-term planning to address problems of migrants’ access to health services. The challenges are not only about the policy content, but also appear in the policy formulation process. The history revealed that none of these policies were formulated according to a rational model where all migrant-related problems were discussed, and all policy options were considered with ample evidence. Previous and current governments did not appear to unpack the structural problems. Oftentimes, policies were quickly generated because of pressures from civil society groups and political tensions.

From the policy implementation angle, this research confirms the importance of a better understanding of implementation processes and the extent to which policy implementation may differ from policy initial intent. The Street-Level Bureaucracy Theory of Lipsky [17] provides a useful perspective in this regard, proposing that frontline implementers have some level of discretion, which enables them to reshape policy for their own ends. Some prior research has also used this theory as their analytic frame, for instance, Walker and Gilson [34] in studying the perceptions of primary care nurses towards user-fee removal policy in South Africa. They reported that primary care nurses were reluctant to grant free services to certain patient groups (despite the policy intent) since nurses considered that many patients were abusing the free care system, and such perceptions were reinforced by the unavailability of medicines at primary care clinics. 

In this study, the zoning policy is an obvious example that is congruent with the Street-Level Bureaucracy theory. The MOI officer admitted that it was impossible to bar undocumented migrants from Myanmar; therefore the zoning policy was a practical measure in their view (although the overarching policy intends to sweep and clean all undocumented migrants). Besides, a reliance on brokers to help migrants pass through all registration process was not uncommon. Some employers of migrants found that a reliance on brokers to facilitate the registration is an effective strategy that helps avoid the bureaucratic hurdles. This discovery further highlights the salience of the concept of street-level bureaucrats in understanding policy. In this case, not only the adaptation of policies by frontline providers during implementation, but also how employers of migrants circumvented the bureaucracy by hiring brokers was noted. These stories clearly broaden the insight of the theory, suggesting that the adaption of policies was done not just by frontline implementers, but also by the users of the policies—in this case, employers of migrants. 

Unclear policy messages and conflicting ministerial policies also aggravated the implementation problems, such as confusion in the cut-off age of dependants and the question of whether the pre-OSS health insurance card was still in effect. These examples are in line with what Schofield mentioned, that implementation failure of a policy can result from various factors including unclear policy messages, insufficient resources, opposition within a policy community, and unfavourable socioeconomic conditions [35]. The incoherence between ministerial policies appeared not only in the implementation details but also in the overarching policy rationale. National security, economic interests, and human rights including health, are all relevant policy drivers of Thai government’s actions. From past till present, it seems that national security and economic interests have usually had the greatest influence on policy decision making, and have dominated health concerns. In other words, state security and economic interests can be seen as ‘high politics’ whereas the health sector was ‘low politics,’ except in special circumstances (for instance, international pressure on Thailand related to the Trafficking Report) [8,36].

This study also highlights challenges faced by most nation states in the modern world—how a country defines ‘social rights’ (including health) for its non-citizens. Harris defined the eligibility to be insured by the HICS amongst undocumented migrants as ‘conditional rights’ [37]. He also argued that Thailand’s extension of insurance coverage for migrants was an example of how a state made judgements with respect to health-related social rights not so much on the basis of nationality or universal personhood alone, but also in conjunction with many other factors, including work status, state security and healthcare costs [37].

Implementation challenges in migrant policies have been observed in not only Thailand, but also other countries. The recent report by the Commission on Migrant and Health strongly underpinned this. While in some countries, like Kenya, despite the existence of national treaties that assert rights to healthcare for all persons, migrants (compared to Kenyan patients) often faced obstacles to care including cost differentials, administrative delays, language and cultural differences, and harassment [38]. A similar situation was found even in some developed nations where healthcare rights of undocumented migrants were relatively generous (as ratified in the byelaws), such as Germany, Italy, Spain, and UK. Gray and van Ginneken [39] reported that some migrants in those countries faced difficulties in accessing healthcare due to various legal interpretations on the scope of services for which undocumented migrants were eligible, not to mention other problems that were not directly related to viewpoints of providers themselves, such as health facilities’ budget constraints and shortage of human resources that are competent in culturally-sensitive care [39,40].

Note that the implementation challenges do not always mean the restriction of healthcare rights of migrants. This can be seen from the recent public debate on migrant data sharing in the UK. In 2017 an MOU between the UK Home Office and National Health Service (NHS) Digital was signed. The MOU formalised data sharing process between the Home Office and NHS Digital and allowed the Home Office to track down migrants upon request on the basis of immigration control [41]. This caused intense public debates in the wider public. The General Medical Council opined that the MOU was rather set out on the grounds of immigration enforcement instead of public wellness. Such debates, amongst other things, resulted in an amendment of the data sharing arrangement between the two agencies. Now the MOU scope is narrowed down, that is, data sharing is permitted only if a person was suspected of serious criminality [38].

There are some policy implications from this study. For short-term recommendations, the government should establish an efficient, transparent, and low cost system for insuring all migrants. Initially, the OSS aimed at perfect registration. Though the OSS is de facto just a facilitating mechanism that requires different ministries to work together in the same venue at the same time, it has not sufficiently tackled the problematic work process and incoherent inter-ministerial policies. Unclear policy messages need to be resolved, and this should be done promptly conditional on mutual agreement between ministries in areas of uncertainty. In addition, a feedback channel that healthcare providers could use to voice their concerns to the MOPH should be established as a matter of urgency. These recommendations will help alleviate the confusion in HICS implementation. Also, healthcare providers may use this channel to consult the MOPH and other central authorities on the extent to which the policies can be tailored or adapted to fit the local context without undermining the overarching policy intention and still maintaining the primary goal in protecting the health of not just migrants, but also all people on Thai soil. 

For long-term recommendations, since migrant health has received much political attention for years (not only in Thailand, but in the whole region), it is vital for the Thai government to extend and strengthen collaboration with its neighbouring countries. Such an approach will likely alleviate many problems originating from the precarious citizenship status of migrants. The OSS and the HICS are in fact temporary measures to tackle health problems of migrants after they crossed the border to Thailand, and while they awaited the NV. Yet, policy initiatives which address the health of migrants before they cross borders are still lacking. In addition, international collaboration should not be just a rhetorical political statement but should be supported by scientific evidence. Thus, further research on effectiveness and feasibility of policies are needed. This should include a study on health system micro-functions at the borders, such as how to ensure a seamless and effective cross-country referral system between hospitals; then broaden the study scope to explore wider health system macro-functions, for instance, the establishment of a cross-border insurance arrangement that can address emerging health needs of people on the move and integration of health information systems between countries. The Association of Southeast Asian Nations (ASEAN) can be the pioneer to advance this agenda. In fact, ASEAN member states have already ratified the rights to health of migrants as evidenced by the Declaration on the Protection and Promotion of the Rights of Migrant workers in 2007 [42]; however, the policy discourse on the Declaration centered on the spread of infectious diseases rather than structural changes in health system functions. 

Inter-governmental collaboration should position health security at the same level as state security and economic benefits. Other supporting mechanisms should be endorsed in parallel to the health insurance initiatives, for example, how to incentivise the recruitment of migrant workers via a lawful channel (like the bilateral MOU). Note that currently there are no additional benefits for a migrant recruited through the MOU, relative to entering the country illegally and awaiting new rounds of registration under the amnesty policies. The greater use of lawful channels might help reduce interference in the recruitment system by private intermediaries or brokers and to some extent help prevent trafficking problems, which are always an irritant for the Thai government.

The bottom line for all recommendations above is protecting the health of migrants, not just selling the insurance card. The OSS and the HICS are merely subcomponents of the whole sphere of migrant policies. A comprehensive and systematic response to migrant health problems is needed. This includes, but is not limited to, policies to minimize illegal border crossing, measures to facilitate lawful hiring of migrants and expedite the NV process, mechanisms to ensure seamless coordination across government authorities, and a meticulous scrutiny of conflicting policy messages.

There remain some limitations in this study. A prime concern was whether and to what extent the findings could be generalized to other settings or to other groups of non-Thai populations. In terms of spatial scope, this study was a case study, where Ranong province was used as a means to investigate how migrant health policies actually functioned in the field. With only one province, where the majority of migrants were Burmese workers, it was difficult to claim that the province was representative of other areas in Thailand.

Secondly, as migrant policies in Thailand are highly dynamic and the timeline for fieldwork quite limited, the study might not be able to capture all the recent changes in migrant policies. For example, the Thai government recently showed signs of expanding the insurance coverage to Vietnamese migrants in order to facilitate free labour movement amongst Southeast Asia nations [43].

Thirdly, the information obtained was a reflection of the respondents’ views, not their exact behaviours. Though the respondents informed the researchers about the adaptation of policies, it was difficult to check if the respondents really behaved in the ways they reported to the researcher.

Fourthly, this paper did not present views of migrants themselves towards the policies. Further research that directly delves into perspectives of migrants would be useful to complement this study. Last but not least, the respondents knew the professional and civil servant status of the lead researcher (RS). This limitation might have shaped the way the informants interacted with the researcher as local health staff might avoid showing negative opinions towards migrant patients to maintain their benevolent image. To address this point, the researchers sought to triangulate interview findings by several means, such as asking for documents that could support the interview findings or re-interviewing some key informants to assess data reliability.

Despite the limitations above, it is hoped that experiences from Thailand can be of value for international audiences. This study highlights the complexity of migrant health and attests that the mere presence of migrant health insurance policy cannot guarantee needed healthcare access amongst migrants—let alone their health outcomes. Protecting the health of migrants requires multisectoral responses and policy coherence from all partners, including public authorities, private sector, international donors, civic organisations, and migrant communities themselves. More importantly, all partners should be aware that health of migrants is not just a matter of the health sector as it is tightly interweaved with various determinants of health, including immigration laws of the recipient countries, employment status, and cultural norms amongst migrants. 

## 5. Conclusions 

Currently, healthcare access of migrants has attracted much attention from all over the globe. Nevertheless, the quest for better protection of migrant health and an extension of UHC to migrants might not be successful if factors surrounding the policy formulation and implementation processes are not sufficiently addressed. The Thai story presented here clearly reflects this concern and may serve as a useful lesson for many countries where the health of migrants is a political and public health issue. The expansion of the HICS benefit package, the instigation of the migrant child insurance in 2013 and the emergence of the OSS clearly reflect how domestic and international politics hugely influenced the design of the policies. At the implementation level, oftentimes, the policies did not function as intended. Ground-level officers could adapt or reshape the policies to address their routine problems. Moreover, the implementation challenges were aggravated by equivocal policy messages and inadequate ministerial coordination. Concerning policy implications, in the short term, the government should resolve policy ambiguities and promote inter-ministerial coordination. In the long-term, the government should strengthen collaboration with neighbouring countries to facilitate the utilization of the MOU channel, which is a lawful cross-border mechanism for migrants; and this will minimize illegal border crossing at the same time. In addition, the health system functions between countries should be harmonized. The feasibility of establishing a cross-border insurance arrangement and seamless cross-border referral system should be further explored.

## Figures and Tables

**Figure 1 ijerph-16-01016-f001:**
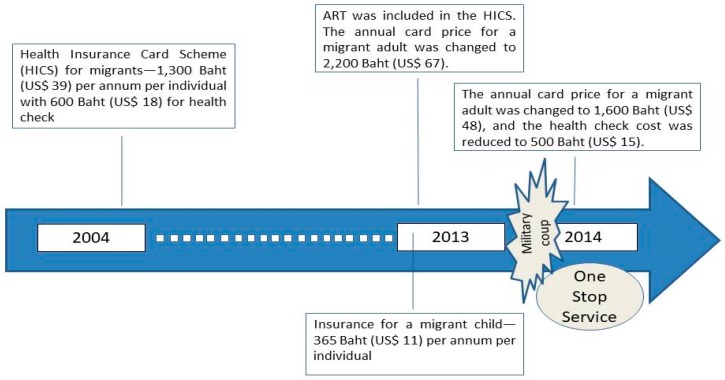
Chronological evolution of the ‘Health Insurance Card Scheme’ for migrants. Source: adapted from Health Insurance Group [11].

**Figure 2 ijerph-16-01016-f002:**
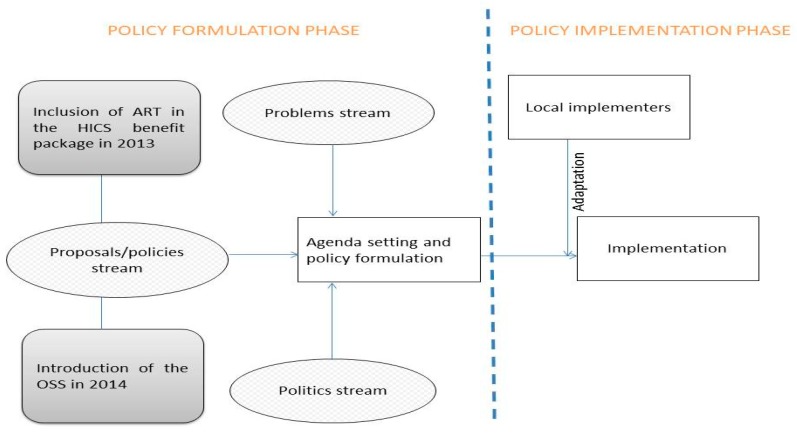
Conceptual framework. Source: adapted from Kingdon [16] and Lipsky [17].

**Figure 3 ijerph-16-01016-f003:**
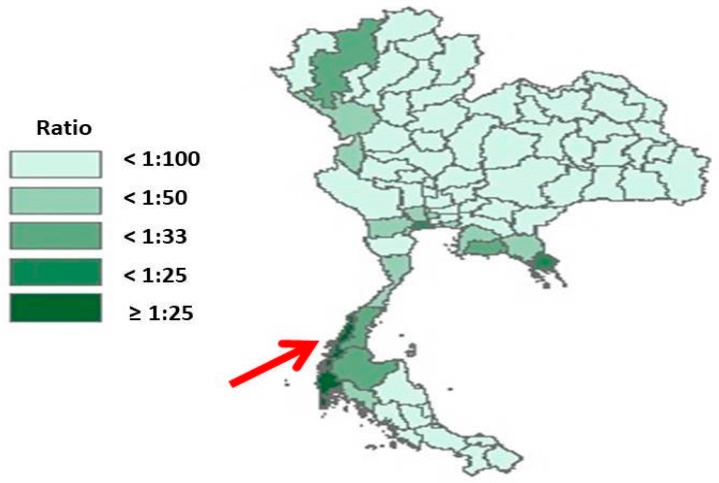
Ratio of registered migrants to Thai citizens in all provinces in Thailand, 2013. Source: adapted from Health Insurance Group, the MOPH [11].

**Figure 4 ijerph-16-01016-f004:**
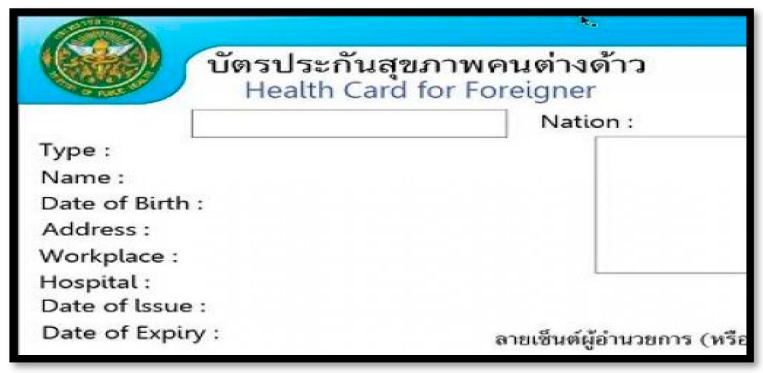
Picture of the health insurance card in 2013. Source: adapted from the MOPH [21].

**Table 1 ijerph-16-01016-t001:** Key characteristics of the participants.

	Code	Current Workplace	Role and Responsibilities
Formulation	PM01	Office of the Permanent Secretary, the MOPH	Policy makers
	PM02	Independent academic institute	Former policy makers
	PM03	Office of the Permanent Secretary, the MOPH	Policy makers
	PM04	Office of the Permanent Secretary, the MOPH	Policy makers
	PM05	Office of the Permanent Secretary, the MOL	Policy makers
	PM06	Faculty of law in one of the public universities	Policy makers
	ADM_CO1	Office of the Permanent Secretary, the MOPH	Administrative staff
Implementation	RN_PHO1	Provincial Public Health Office	Administrative staff
	RN_PHO2	Provincial Public Health Office	Executive staff
	RN_RNH1	Provincial Hospital X	Executive staff
	RN_RNH2	Provincial Hospital X	Insurance staff
	RN_RNH3	Provincial Hospital X	General practitioner
	RN_KH1	District Hospital Y	Insurance staff
	RN_KH2	District Hospital Y	Executive staff
	RN_NGO1	Foundation A	NGO
	RN_NGO2	Foundation B	NGO
	RN_HC1	Health centre A	Executive staff
	RN_HC2 *	Health centre B	Executive staff
	RN_HP1	Health centre B	Village health volunteer
	RN_MOI1 *	Department of Provincial Administration, the MOI	Executive staff
	RN_WP1	Provincial Employment Office, the MOL	Executive staff
	RN_E1	Construction site A	Employer of migrants
	RN_E2	Rubber field A	Employer of migrants
	RN_E3	Fish dock A	Employer of migrants
	RN_B1 *	Fish dock B	Employer of migrants

Note: MOI = Ministry of Interior, MOL = Ministry of Labour, MOPH = Ministry of Public Health, NGO = Non-government organisation, * = telephone interview.

**Table 2 ijerph-16-01016-t002:** Contrasting policy intention and policy implementation.

Time	Policy Intention	Policy Implementation	Examples of Interviewees Raising This Point
Pre-OSS (before 2014)	Expansion of benefit package to cover HIV/AIDS treatment	The number of migrants enrolled in insurance decreased substantially as the premium increased.	PM03, RN_RNH1, RN_E2, and RN_E3
	Allowing all migrants regardless of their work status to buy insurance	Some Europeans with chronic diseases came to buy insurance at health facilities.	PM_03
Post-OSS (after 2014)	Finishing the nationality verification by late March 2015	Many migrants (precise number unknown) failed to participate in nationality verification.	RN_B1, RN_E1, RN_E2, and RN_E3
	Aiming to cover children of migrants	Unclear policy message regarding the definition of dependents made some providers reluctant to sell insurance to migrants’ children.	RN_RNH2 and RN_RNH3
	Smooth-running migrant registration process.	Some employers paid private intermediaries to help overcome red tape in registration.	RN_B1 and RN_E2
	Integration of inter-ministerial functions.	Lack of feedback mechanism for concerns at ground level to reach central authorities.	RN_PHO1, RN_PHO2, RN_RNH1, RN_RNH2, RN_MOI1, and RN_WP1

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
