# Peer review of "The Devil Is in the Detail—Understanding Divergence between Intention and Implementation of Health Policy for Undocumented Migrants in Thailand"

_ijerph, 2019, doi:10.3390/ijerph16061016_

Round 1

Reviewer 1 Report

Thank you for the opportunity to review this very interesting article. I applaud your efforts to describe a very nuanced situation in accessing healthcare in a politically charged environment for a very vulnerable population of migrants.

Comments:

Abstract

L22 What type of framework analysis? Perhaps list them and that this is deductive and inductive. Also, why did you choose Ranong? There are clear reasons, perhaps mention them briefly here.

L30-1 You end the abstract and the paper with "minimize illegal border crossing". That doesn’t seem to be the main points of the article, which documents the challenges of a nation trying to secure health care coverage for immigrants, legal or otherwise.

Intro

L48-9 3 million immigrants over what time period (this comes up again in lines 71-3)

L64 choose another word other than “demeaning”

Figure 1

Include $US here too, as you did in the text

Figure 2

At least in my printed copy, there were some formatting issues in the proposals/policies stream bubble

Figure 3

Highlight where Ranong is with an arrow. I can guess, but don’t make the reader guess.

Table 1

With this much information about respondents I might be able to track down who they were. I’m not sure it matters what the specific ages or genders are of each respondent (especially as you attribute specific quotes to each). Perhaps just give a summary of age/gender and examples of workplaces and roles/responsibilities? You could use the role/responsibilities in lieu of your codes in the results when you attribute comments and quotes to individuals.

Methods

L174-6 You repeat your IRB approval here when you stated it in section 2.3.

Results

L210 Please spell out TIPS or introduce the acronym beforehand. I looked back to find it and wasn’t immediately successful. Your readers won’t be either.

L221-3 The respondent suggests a systematic response and I was left wondering what it might be. Perhaps refer back to this in the discussion?

L233-4 You suggest helping neighbors to develop insurance systems… through what mechanism? What would this mean for emergent care/daily needs?

L253 and 303 Fix spacing issues between the text  and your headers.

L329-331 This quote left me wondering again. Perhaps include a general statement about what it means that this respondent thinks the Thai people need to petition? (without putting words in respondent’s mouths… does this link back to discussion again?)

Discussion

L443-449 The first lines of the discussion are worth highlighting in the abstract. I think this and the challenges of accomplishing this are much more relevant as closing points to both the abstract and the conclusion.

L519-522 While it's from another important implementation theorist, the Schofield citation could use a bit more linking to the content of your results.

Overall

Excellent work and well written. I look forward to seeing it in print.

Reviewer 2 Report

This is a well-written article highlighting the gaps and dissonance between policy and implementation in the context of health policies for undocumented migrants in Thailand. This issue is also very important as Thailand will become a full-fledged aging society by 2021 and migrant workers are necessary for sustaining the society. At global level, such a policy-implementation gap is also a major concern in many countries, and the lessons learned from this study will be useful for other countries where migrant work is common. The analysis was well done using two theories (Kingdon’s Agenda Setting Theory and Lipsky’s Street-Level Bureaucracy Theory) to answer the research questions. However, the article can be more improved by overcoming some issues described below.

1.      L 22, L113, L178. ‘Framework analysis’ and ‘policy analysis’ are used interchangeably? You may add references to support it.

2.      L93 Fig 1: It depends on journal policy but usually figures which are not original of the authors should not be used. If it can be used, US$ should be added, though exchange rate information is clearly written within the main text.

3.      L105-110: These statements may rather fit to the result section as they show inclusion and exclusion criteria.

4.      L 106,L141: The authors mentioned that the population scope was limited to Cambodian, Laotian, and Myanmar migrant workers. However, the setting was in Ranong province, which borders Myanmar and the majority of the workers there come from Myanmar. Is data available about the (estimated) number of Cambodian and Laotian migrant workers in Ranong province? If the authors could provide more data on the number of migrant workers there, the argument would be stronger. If the overwhelming majority of workers in Ranong province is Myanmar workers, it might be better to rewrite the population scope. The difficulties in engaging with Myanmar workers might be different from that of Laotian (due to mutual-intelligibility of Thai and Lao) or Cambodian workers.

5.      L113: Not many, but research on policy-implementation gap has been conducted for different health topics. It is optional, but I wonder if the authors can justify the selection of this research method.

6.      L131: Suppose this method is well justified, Figure 2 could be made easier to understand and more informative. For example, the authors could create the conceptual framework vertically and distinguish between the policy makers and the implementers.

7.      L159: Again, it depends on journal policy, but showing exact number of age might have a risk of identifying the person as the number of VIPs or key persons might be limited. Sometimes another expression is used such as 50s or 60s.

8.      L181: For qualitative research, there are many ways to overcome validity threats. In this study, only member checking was used for it?

9.      L184: As the authors aimed to highlight the gaps and dissonance between objectives and implementation of health policies, the themes from the framework analysis could be presented in a way that is easier to understand. I believe the main aim is to impact change and the only way is to get the policy makers to read this important piece of work. For example, creating a table where the first column is the themes from the qualitative analysis, second column is the supporting quotes from the policy makers, third column is the supporting quote from the implementers, and the fourth column is the authors’ interpretation of the gaps and dissonance between the two, will be much easier to understand.

10.  L225 and other parts: Some data (?) written in the results section seem to be known before the interview. I wonder what should be written in the background and what should be written in the results. If some of these data (?) in the results are the ‘results’ of the study, data source might be interview plus something different such as published and non-published document of MOPH.

11.  L252: To date means when?

12.  L269: How about Asians, such as Japanese, Korean, Chinese, Singaporean? Are they minority foreign workers?

13: L443: The paragraph sounds like more justification of this study and may fit well in the introduction.

14: L450: ‘Devil in the detail’ is reflecting the contents of this study very well and is potentially an attractive phrase for the title.

15: L455: “It is clear” or “It suggests”?

16: L466: Ref 33 was published in 1957. Any new references to support that it is still true even in these days.

17: L442: Discussion is well written and persuasive, but it seems to be too long. I wonder if it can be shorten highlighting the new finding of this study.

18: L624: Out of 42 references, only 11 were academic ones published in peer-review journals. I wonder if more academic references can be used to strengthen the contents of this study.

Reviewer 3 Report

It is a well written article about an interesting public health item.

The text is rather long, especially the results and discussion section. It will be more accessible for an outsider if the text will be shortened.

Figure 3 is of poor quality.

Table 1 has a lot of details and can be simplified. It is questionable if these details respect sufficiently the individual privacy of respondents.

The recommendation in the abstract and conclusion about 'minimize illegal border crossing' is a political one, not justified by the research. From a public health perspective, is seems more justified to harmonize the health systems on micro and macro level between countries (discussion section 554-558).
